# The Influence of Long-Term Different Crop Rotations and Monoculture on Weed Prevalence and Weed Seed Content in the Soil

Lina Marija Butkevičienė [1], Lina Skinulienė [1,*], Ingė Auželienė [2], Vaclovas Bogužas [1], Rita Pupalienė [1] and Vaida Steponavičienė [1]





[1] Institute of Agroecosystem and Soil Sciences, Vytautas Magnus University, Studentu Street 11, Akademija, LT-53361 Kaunas, Lithuania; lina.butkeviciene@vdu.lt (L.M.B.); vaclovas.boguzas@vdu.lt (V.B.); rita.pupaliene@vdu.lt (R.P.); vaida.steponaviciene@vdu.lt (V.S.)
[2] Department of Landscape Design and Recreation, Kaunas Forestry and Environmental Engineering University of Applied Sciences, Liepų Street 1, Girionys, LT-53101 Kaunas, Lithuania; inge.auz@gmail.com
* Correspondence: lina.skinuliene@vdu.lt

**Abstract:** Equally effective way to achieve sustainable farming and the challenge set by the European Commission on 20 May 2020: proper crop rotation and thus reduction of the quantity of on-farm chemicals. Long-term stationary field experiments were established in 1966 at Vytautas Magnus University Experimental Station ($54°53'$ N, $23°50'$ E). The study was conducted with intensive, three-course, field rotation with row crops, for green manure crop rotations, and rye monoculture as well during the last 5-year period of a 50-year investigation to determine the effect of crop rotation combinations and rye monoculture on weed density and seed bank and grain yield. In cereal crops, weed counting was performed twice: weed density was determined before the application of herbicides, and weed counting was done before the harvest. Weed seedlings were counted, their botanical species were determined, annual and perennial weed number was estimated. Weed seed bank was established before primary tillage in soil. The results obtained confirmed the hypothesis that with climate change and intensive farming, long-term crop rotations are likely to increase crop productivity, reduce weeds and weed seed banks in the soil, and thus contribute to maintaining agroecosystem sustainability. The winter rye 1000 grain weight and yield decreases as weed mass increases showing strong negative correlations: $y = 475.56 - 11.93x$, $r = -0.91$, $p \leq 0.05$; $y = 82.97 - 14.82x$, $r = -0.97$, $p \leq 0.01$. Reseeding of rye crops leads to a growing prevalence of weeds such as *Equisetum arvense* L. and *Mentha arvensis*. Crop structures these days are dominated by cereals, which inevitably increase the spread of weeds, and therefore, the importance of crop rotations increases in the context of intensive farming.

**Keywords:** long-term rye monoculture; pre-crops; annual weeds; perennial weeds; grain yield

## 1. Introduction

Weeds are plants that grow in crops and stunt agricultural crops, reducing the yields and degrading their quality. They are one of the biggest risks to crop productivity, which could lead to a loss of about 34% of yields worldwide [1]. Savary et al. [2] point out that weeds are greater reducers of crop yields and in many cases are more economically important than insects, fungi, or other harmful organisms. Swedish scientists have conducted large-scale, long-term (26 years) research and found that crop losses due to the abundance of weed biomass are as high as 31%, even if herbicides are used. It has also been shown that the use of herbicides changes the species composition of weeds, and the loss of yield due to weeds is almost always caused by the predominance of different weed species in the crop [3]. Researchers recognize that the use of herbicides is the most successful, profitable, and beneficial means of weed control [4,5]. Unfortunately, over-reliance on herbicides has

led to resistant biotypes of weeds [6,7], crop phytotoxicity [8], environmental pollution, and risks to public health [9]. However, weeds are natural components of cultivated plant communities that maintain ecological balance in the crop, but only if they do not exceed the degree of harmfulness [10,11]. The current herbicide-based weed control model is generally considered unsustainable. Strict EU directives limit the number of herbicide sprays in crops and increase the risk of herbicide resistance [12]. Physical methods of weed control are generally less effective than herbicides, and successful control of weeds in crops often requires preventive tactics such as catch crops. They play a key role in increasing the efficiency of land use, weed control, improving environmental performance, and increasing economic profitability [13–15]. The main elements of sustainable agricultural activity are crop rotation with legumes, roots, and catch crops, as well as tillage using a mulch of plant residue. The conditions of the crop rotation system not only ensure a high and stable yield but also enrich the soil with organic carbon [16]. The timing of weed control is also important: eradication, both chemical and mechanical, must be initiated while weeds are small, at the stages of seedling, and sometimes properly re-treated after a short interval for proper control [13,15]. Poor crop competitiveness due to limited nutrient content, insufficient knowledge of perennial weeds among growers, and frequent application of various mechanical effects are considered to be the main reasons for the intensity of perennial weeds [17]. Weed control in crop rotations can be successful when agricultural crops with optimal vegetation periods and competitive characteristics are selected [18]. Winter rye (*Secale cereale* L.) is most effective at putting weeds into the shade compared to other Poaceae cereals. They spread and strengthen at the end of autumn vegetation and cover the soil surface well. Swedish researchers Milber and Hallgren [3] also indicated that the crops least affected by weeds were spring barley (*Hordeum vulgare* L.), spring wheat (*Triticum aestivum* L.), oats (*Avena sativa* L.), winter wheat, and rye. Utilizing the resilience of the crop can greatly reduce the cost of weed control. *Poaceae* are more resistant to weeds than beans, but the resisting power of *Poaceae* and Fabaceae mixtures is even higher compared to a homogeneous crop [19,20]. Scandinavian researchers argue that the prevalence of *Elytrigia repens* (L.) Nevski in crop rotations was reduced 28% with catch crops and manure application. In potatoes (*Solanum tuberosum* L.) and winter rye, the population of *Elytrigia repens* (L.) Nevski also increased but inter-row loosening reduced their population. Thus, there are crops that need special attention when creating crop sequences. To ensure weed control, it is necessary to ensure sufficient nutrients, and to properly control the spread of *Elytrigia repens* (L.) Nevski, mechanical interventions are needed [17].

Empirical studies of weed behavior in long-term crop rotations allow determining the interaction of system components and the effect of pre-crop (the previous crop) on the plant of the next vegetation period. However, these aspects are complex and difficult to control due to environmental effects. For example, a slower nutrient release from organic residues and fertilizers compared to synthetic fertilizers can alter the competition of crops and weeds for nutrients (especially nitrogen) and clearly affect the next season's plant development as well as competition with weeds.

Short crop rotations applied by farmers often do not play a role in crop rotation. There is a lack of knowledge about the effects of different combinations of crop rotations on mist yields, elements of crop structure, and crop weediness under current farming and changing climate conditions.

Research hypothesis: with climate change and intensive farming, long-term crop rotations are likely to increase crop productivity, reduce weeds, and weed seed bank in the soil compared to monoculture, and thus contribute to maintaining agroecosystem sustainability. The research objective is to determine the effect of crop rotation combinations and rye monoculture on weed density and weed seed bank and grain yield after 50 years of investigation.

## 2. Material and Methods

Prevalence of weed assessment was conducted in long-term crop rotation which was initiated in 1966 at Vytautas Magnus University Experimental Station in Lithuania (54°53′ N, 23°50′ E) and has been continued until now. All crop rotations (Table 1) were arranged in time and space each year, three replications were applied. Each main plot was 18 m long by 9.60 m wide. The investigation was performed on winter rye (*Secale cereale* L.) 'Matador'.

**Table 1.** Crop rotation sequences.

| Crop Rotation | Crop Rotation Components |
|---|---|
| Field rotation with row crops | (1) Winter wheat (*Triticum aestivum* L.) + undersow; (2) Perennial grasses (*Trifolium pratense* L. + *Phleum pratense* L.) used first year; (3) Perennial grasses used second year; (4) Winter rye (*Secale cereale* L.); (5) Sugar beet (*Beta vulgaris* L.); (6) Spring barley (*Hordeum vulgare* L.); (7) Oat (*Avena sativa* L.); (8) Black fallow. |
| Intensive | (1) Vetch-oat (*Vicia sativa* L. + *Avena sativa* L.) mixture for fodder + undersow; (2) Perennial grasses (*Trifolium pratense* L. + *Phleum pratense* L.) used first year; (3) Winter rye (*Secale cereale* L.) and after intermediate crop, winter rape (*Brassica napus* L.); (4) Potatoes (*Solanum tuberosum* L.) and after intermediate crop, winter rye (*Secale cereale* L.) for fodder; (5) Corn (*Zea mays* L.); (6) Sring barley (*Hordeum vulgare* L.) and after intermediate crop, oil radishes (*Raphanus sativus* L.). |
| For green manure | (1) Lupines (*Lupinus angustifolius* L.) for green manure; (2) Winter rye (*Secale cereale* L.); (3) Winter rape (*Brassica napus* L.) for green manure; (5) Winter rye; (6) Potatoes (*Solanum tuberosum* L.); (7) Spring barley (*Hordeum vulgare* L.). |
| Three-course | (1) Black fallow; (2) Winter rye (*Secale cereale* L.); (3) Oat (*Avena sativa* L.). |
| Winter rye monoculture | (1) Winter rye (*Secale cereale* L.). |

The study was conducted in 4 different crop rotations: intensive, three-course, field rotation with row crops, for green manure and rye monoculture as well. The study was conducted in the year 2012–2016.

The soil of the experimental site was *Endocalcari-Epihypogleyic Cambisol* (sicco) (CMg-p-w-can) [21].

Granulometric composition—dusty loam on loam and clay. Average nutrient contents in the soil were analyzed (data for 2011, 2014, and 2016) at 0–20 cm depth, samples were taken from 20 sites using an agrochemical drill in every treatment: $pH_{KCl}$—from 6.6 to 7.0; $P_2O_5$—from 131 to 206.7 mg kg$^{-1}$; $K_2O$—from 72.0 to 126.9 mg kg$^{-1}$. $P_2O_5$ and $K_2O$ were analyzed by the Egner-Riehm-Domingo (AL) method [22]. $C_{org}$ from 11.3 to 15.6 g kg$^{-1}$ by the spektrofotometrical method.

During the experiment, the same soil tillage system was implemented, and plant protection products were used as needed.

### 2.1. Experimental Agrotechnics

Until 1990, all crop rotations were fertilized in such a way that during this period, each crop rotation would receive the same amount of mineral fertilizers, and it was then fertilized at normal rates suitable for agricultural crops. In the experiment conducted in 2012 and 2016, tillage was carried out according to common cultivation technologies of winter rye cereals. After the first cut, perennial grasses were disked in the field crop rotation with row crops, and intensive crop rotations. In early August, winter rape for green manure was disked in the green manure crop rotation. Plowing in all crop rotations was performed 10–15 days after the rye harvest. Straw was retained and incorporated as an organic fertilizer (Table 2). In early September, the soil was cultivated twice, $N_8P_{20}K_{30}$ 300 kg ha$^{-1}$ was applied before the first cultivation. Winter rye cv. "Matador" (180 kg ha$^{-1}$) was sown. At the beginning of the spring vegetation of winter rye, the plots were fertilized with 200 kg ha$^{-1}$ ammonium sulfate ($N_{26}S_{13}$), and additionally fertilized with ammonium nitrate ($N_{34}$) 250 kg ha$^{-1}$ after two weeks. Crop stands were sprayed with growth regu-

lators Cycocel 750 SL1 at 1.2 L ha$^{-1}$ (a.i. chlormequat chloride 750 g L$^{-1}$) and Stabilan 750 SL (a.i. chlormequat chloride 750 g L$^{-1}$). In spring, winter rye was sprayed with the herbicide Arelon flussig at 1.2 L ha$^{-1}$ (a.i. isoproturon 50 g L$^{-1}$) 2.0 L ha$^{-1}$, 1.0 L ha$^{-1}$, with fungicides INPUT 460 EC (a.i. prothioconazole 160 g L$^{-1}$, spiroxsamine 300 g L$^{-1}$) 1.0 L ha$^{-1}$ and with Fandango (a.i. prothioconazole 100 g L$^{-1}$, fluoxastrobin 100 g L$^{-1}$) 1.0 L ha$^{-1}$.

**Table 2.** Sources of organic matter in crop rotations.

| Crop Rotations | Crops | Organic Matter Source | | | |
|---|---|---|---|---|---|
| | | Manure (55 Mg ha$^{-1}$) | Straw | Green Manure | Perennial Grasses |
| Field with row crops | Winter rye | + | + | | + |
| Intensive | Winter rye | + | + | + | + |
| For green manure | Winter rye | | + | + | |
| Three-course | Winter rye | | + | | |
| Rye monoculture | Winter rye | | + | | |

+ indicates the Organic matter source in a particular crop rotation.

### 2.2. Meteorological Conditions

Warm and humid weather prevailed during the sowing of winter cereals in September 2011. The temperature was 1.4 °C and the precipitation exceeded the long-term average by 20.1 mm. Meteorological indicators in October and November were close to the long-term averages. Consequently, the period of preparation for wintering was favorable for cereals to bush (Tables 3 and 4).

**Table 3.** Average temperature (°C) and the sum of the active temperatures (SAT) during the winter rye growing season (September–August) in 2011–2016, Kaunas Meteorological Station.

| Year/Month | 09 | 10 | 11 | 12 | 01 | 02 | 03 | 04 | 05 | 06 | 07 | 08 | SAT |
|---|---|---|---|---|---|---|---|---|---|---|---|---|---|
| 2011–2012 | 13.6 | 7.4 | 3.5 | 1.9 | −2.7 | −8.8 | 1.8 | 7.7 | 13.7 | 15.3 | 19.4 | 17.2 | 2555.4 |
| 2012–2013 | 13.3 | 7.6 | 4.8 | −4.4 | −6.9 | −1.0 | 3.9 | 5.5 | 16.6 | 18.5 | 19.2 | 18.4 | 2746.9 |
| 2013–2014 | 12.3 | 9.0 | 5.0 | 2.1 | 5.0 | 0.7 | 5.1 | 9.1 | 13.2 | 14.6 | 20.5 | 17.7 | 2641.1 |
| 2014–2015 | 13.5 | 7.9 | 2.8 | −0.8 | −0.4 | 0.2 | 4.6 | 7.1 | 11.4 | 15.4 | 17.4 | 20.3 | 2490.5 |
| 2015–2016 | 14.3 | 6.2 | 4.9 | 2.6 | −7.1 | 0.6 | 2.1 | 7.4 | 15.7 | 17.2 | 17.9 | 16.9 | 2544.7 |
| Long-term average 1974–2013 | 12.6 | 6.8 | 2.8 | −2.8 | −3.7 | −4.7 | 0.3 | 6.9 | 13.2 | 16.1 | 18.7 | 17.3 | - |

SAT = sum of active temperatures (≥10 °C).

**Table 4.** Precipitation (mm) during the winter rye growing season (September–August) in 2011–2016, Kaunas Meteorological Station.

| Year/Month | 09 | 10 | 11 | 12 | 01 | 02 | 03 | 04 | 05 | 06 | 07 | 08 | Sum |
|---|---|---|---|---|---|---|---|---|---|---|---|---|---|
| 2011–2012 | 73.9 | 21.6 | 15.5 | 47.5 | 47.6 | 33.8 | 16.2 | 72.3 | 50.3 | 93.4 | 112.8 | 69.2 | 654.1 |
| 2012–2013 | 67.2 | 75.0 | 68.7 | 43.3 | 47.2 | 40.7 | 9.5 | 56.5 | 63.8 | 45.9 | 118.5 | 67.2 | 752.7 |
| 2013–2014 | 104.3 | 43.7 | 63.1 | 36.4 | 53.2 | 26.6 | 29.6 | 21.3 | 84.2 | 49.4 | 52.5 | 111.3 | 675.6 |
| 2014–2015 | 20.7 | 93.4 | 29.5 | 49.3 | 74.4 | 12.8 | 45.7 | 46.0 | 43.8 | 16.4 | 72.4 | 6.9 | 511.3 |
| 2015–2016 | 56.6 | 18.2 | 95.6 | 61.3 | 41.6 | 68.4 | 47.2 | 41.2 | 36.4 | 83.9 | 162.9 | 114.9 | 828.2 |
| Long-term average 1974–2013 | 60.0 | 51.0 | 51.0 | 41.9 | 38.1 | 35.1 | 37.2 | 41.3 | 61.7 | 76.9 | 96.6 | 88.9 | 679.7 |

Based on the amount of precipitation, 2012 can be considered a humid year, especially the spring period. Precipitation was high in April (72 mm) and June (92 mm) compared to the long-term averages of 41.3 mm and 62.9 mm, respectively. These were particularly favorable conditions for the weeds to spread. July was exceptionally dry, with only 12 mm of precipitation which was significantly lower than the long-term average of 96.6 mm. The average monthly temperature during this period was 2 °C higher than the long-term average, reaching 17.8 °C. In the following months of August and September, precipitation was close to the long-term average. It was very high in October (75 mm) and exceeded the long-term average for this month by 1.5 times. The temperature was close to the long-term average. In November, the temperature was 1.3 °C higher than the average, making the autumn of 2012 humid and warm.

Spring of 2014 could be distinguished by heat and a lack of humidity. The spring vegetation of cereal started particularly early in March. In the second decade, the average daily temperature was 4.1 °C, and in the third, it was 8.0 °C. This month's average temperature exceeded the long-term average by 4.8 °C (Table 3). April was also 2.2 °C warmer than usual and the precipitation was 20.0 mm lower (Table 4). Such high temperatures in early spring are not typical of the Lithuanian climate. Only in the third decade of May was significant precipitation observed, but both June and July were dry. In July, the temperature exceeded the long-term average by 2.9 °C. The temperature in autumn (September, October, and November) was typical for this period. In September, 13.5 °C prevailed, but the precipitation was almost 3.0 times lower than the long-term average. In October, precipitation was 31.3 mm higher than usual, while November was less humid than usual. The average daily temperature in spring and summer 2015 did not significantly differ from the long-term average, but this year could be distinguished by a lack of precipitation. Precipitation in March and April was still typical for this period, but in May it had already decreased by 17.9 mm. June was particularly dry with only 16.4 mm, which was 60.5 mm less than the long-term average of this month. In July, precipitation was also 24.2 mm lower than typical. During the sowing of cereal in September, meteorological conditions were favorable for their germination, but the dry period re-emerged in October with 32.8 mm lower precipitation compared to the long-term average.

The conditions for the beginning of winter cereal vegetation were very favorable in spring 2016. Already in the first decade of April, the average daily temperature rose to 8.4 °C and humidity was typical for this month. In May, warm and dry weather prevailed. The average daily temperature was 2.5 °C higher and the precipitation was 1.7 times diminish than the long-term average. Precipitation was only higher in the second decade of June. The average daily temperature was also 3.0 °C higher than usual. July started with heavy rains that lasted all month. This month, the average daily temperature exceeded the long-term average by 2.0 °C and the precipitation was almost twice. August was exceptionally humid with the precipitation being 1.4 times higher than the long-term average. This prolonged rainy period created particularly unfavorable conditions for normal grain ripening and harvesting.

### 2.3. Weeds and Species Composition of Dominant Weeds

In cereal crops, weed counting was performed twice. For the first time, BBCH 21 weed density was determined before the application of herbicides. Ten areas were randomly selected in each plot, using 20 cm × 30 cm (0.06 m$^2$) wire frames, covering two lines of cereals in the plots. Weed seedlings were counted, their botanical species were determined, annual and perennial weed number was estimated. The resulting number of weeds was converted to the number per m$^{-2}$. For the second time, weed counting was done before the harvest, before BBCH 90, in 10 randomly selected areas in each plot. By placing the frames across two rows of cereal, the weeds in the frame area were plucked and wrapped in paper. After drying, the botanical composition of the species was analyzed. Weeds of each species were counted and weighed. The results were converted to the number per m$^{-2}$ and the weed dry mass—to grams per m$^{-2}$.

### 2.4. Weed Seedbank

The weed seed bank was established before primary tillage in the soil at depths of 0–10 and 10–20 cm, samples were taken from each experimental plot (from all replications) from 20 places using an agrochemical drill (Ø 20 mm), in 2012, 2014, and 2016, an average sample was compounded. Two weighed samples of 100 g of dry soil were placed in a 0.25 mm sieve and washed with a stream of running water until small particles of soil were washed away. Weed seeds and the remaining mineral particles of soil were separated from the organic residues by a saturated saline solution [23].

The separated weed seeds with the saline solution were poured onto filter paper, washed with clean water, and dried. The quantity of weed seeds found in the sample was converted into a unit of area ($m^{-2}$), as well the species composition of the seeds was determined.

$$A = n \times h \times q \times 100 \tag{1}$$

A—number of weed seeds per $m^{-2}$; n—number of seeds in the sample; h—thickness of the examined arable soil layer in cm; q—soil density $g\ m^{-3}$.

### 2.5. Statistical Analysis of the Experimental Data

The statistical evaluation of the quantity of weed seedlings presented in the article was performed by taking the year as a replication. The quantity and weight of weeds, as well as the quantity of weed seeds at a depth of 0–20 cm, were statistically estimated every year. Winter rye yield was assessed every year. Data were processed using a one-factor ANOVA from the software package SYSTAT 10 [24,25]. The significance of differences in all treatments was assessed by the LSD test. The data of weed prevalence and abundance that did not conform with the normal distribution law were transformed using the Log (X) function before statistical processing. The relationship between the variables was assessed by correlation and regression using statistical data analysis software STAT from the computer software package SELEKCIJA [26,27].

## 3. Results and Discussion

In 2012–2016, the number of weed seedlings in rye monoculture in spring was on average 2.7 times higher compared to rye crops in different crop rotations (Figure 1). During the study period, two to three times fewer weed seedlings were found in the three-course crop rotation compared to other crop rotations and 4.8 times less compared to the monoculture. Weed germination in a field with row crops did not differ significantly compared to a three-course crop rotation of rye. A short crop rotation and black fallow every three years reduced weed germination in winter rye crop compared to long crop rotations and monoculture. Storing black fallow in a three-course crop rotation also reduced the spread of perennial weeds. Compared to three-course crop rotations, more weeds were found in other crop rotations: intensive crop rotation—42.8%, green manure rotation—51.5%, and in monoculture—79.6% more weeds than in three-course crop rotation. No significant differences in the prevalence of weed seedlings in crop rotations were found because of the high dispersion of data. The results showed that the different meteorological conditions of each year did not affect the prevalence of weed seedlings, the amount of weed seedling was the highest in rye monoculture.

It has been established that the weeds that have survived until the end of July make up only half of the total number of those that germinated in spring [28]. In 2012–2016, before the harvest, 37 weed species belonging to 17 families were found in the crops of winter rye grown in five crop rotations: *Apiaceae, Asteraceae, Equisetacea, Brassicaceae, Amaranthaceae, Caryophyllaceae, Euphorbiaceae, Fabaceae, Juncaceae, Lamiaceae, Poaceae, Plantaginaceae, Polygonaceae, Ranunculaceae, Rosaceae, Rubiaceae, Violaceae.*

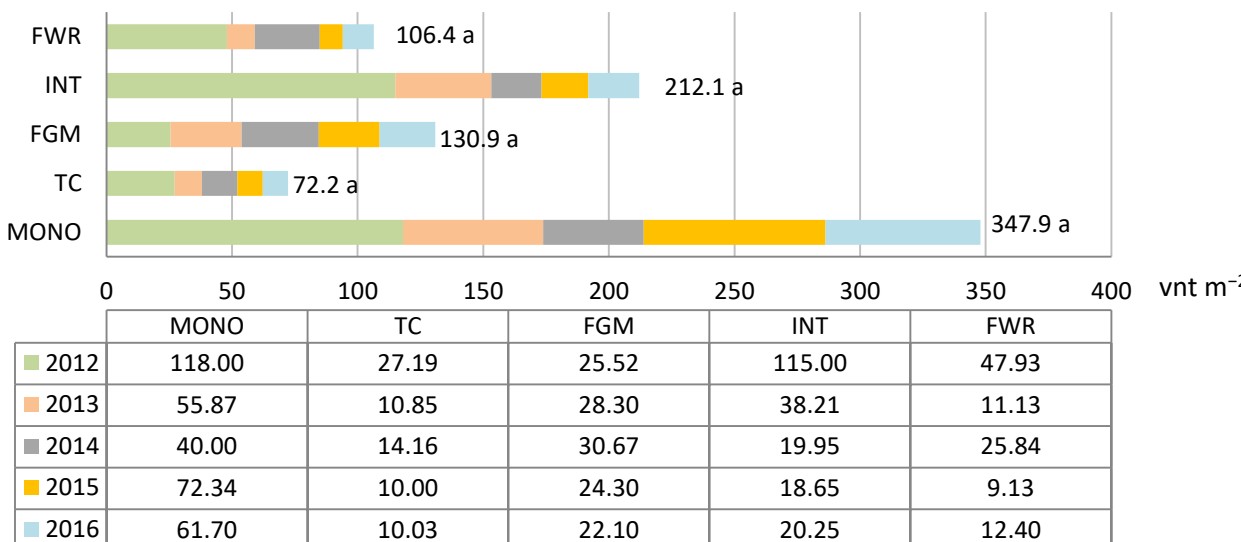

**Figure 1.** The amount and 5 years average of weeds seedlings per m$^{-2}$ in winter rye crops, 2012–2016. Note: different letters indicate significant differences between the treatments ($p \geq 0.05$); FWR—field rotation with row crops; INT—intensive rotation; FGM—rotation for green manure; TC—three-course rotation; MONO—rye monoculture.

Short-lived weeds were only more widespread in the humid year of 2012. Moisture-loving weeds (*Echinochloa crus-galli* (L.) P. Beauv., *Poa annua* L., *Juncus bufonius* L.) were particularly spread in intensive crop rotation (Figure 2). Plant residues of catch crops may have had an impact, as they retained moisture in the soil better in wetter years, thus creating favorable conditions for the spread of moisture-loving weeds. In monoculture, the dominating annual weeds were *Tripleurospermum perforatum* (Mérat) M. Laínz, which accounted for 34.4% of the total number of annual weeds in this crop rotation. *Apera spica-venti* P. Beauv. accounted for 23.4%, and *Consolida regalis* Gray for another 20.3%. The latter weed was only found in monoculture. *C. regalis* did not spread in other crop rotations and was not observed in all years of the experiment. Application of winter and summer crop rotation, while maintaining crop biodiversity, limits the multiplication and spread of weeds of different biological groups [29–31].

Monocotyledonous weeds were widespread in the field crop rotation. Most of the annual weeds were *Apera spica-venti* P. Beauv. *Poa annua* L. was also found in this crop rotation. It accounted for 35.3% of the total number of annual weeds. *E. crus-galli* accounted for 71.4% of the number of annual weeds in the crop rotation for green manure.

In three-course crop rotation, 25.0% of all annual weeds were *Stellaria media* L. and 16.6%—*Galinsoga parviflora* Cav. *P. annua* (46.7%) and *Galium aparine* L. (18.4%). In intensive crop rotation, 6 species of annual weeds, evenly distributed according to the number per m$^{-2}$, were identified. *Capsella bursa-pastoris* (L.) Medik., *E. crus-galli*, *P. annua*, and *Persicaria lapathifolia* (L.) Gray were found among them.

The year 2012 was also favorable for the spread of perennial weeds (Figure 2). *M. arvensis* (57.9%) and *E. arvense* (26.3%) dominated in monoculture, but *E. arvense* accounted for the largest share in terms of weed dry mass, Since perennial rhizomes and root weeds are the most harmful [31,32], it can be stated that the problem of weed spread is even difficult to solve in monoculture with the use of herbicides. In Poland, in 24-year-old cereal monoculture, the number and the biomass of weeds were found to have significantly increased in the first year of monoculture, followed by stabilization in the following years. In crop rotations, the number and the biomass of weeds were significantly lower compared to monoculture and depended on previous crops, but the main weed species belonging to the families of *Polygonaceae* and *Chenopodiaceae* dominated [33]. In 2013, perennial weeds *Elytrigia repens* (L.) Nevski, and *Sonchus asper* L. Hill. increased in fields with green manure crop rotation.

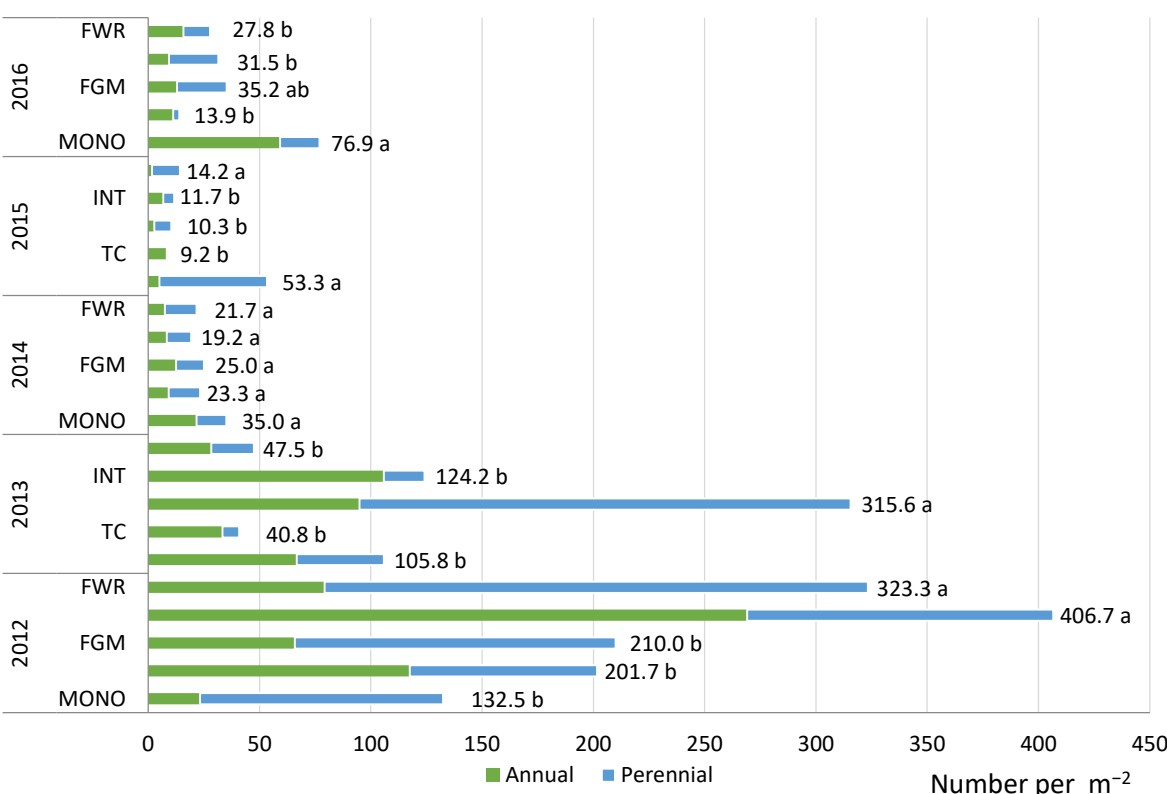

**Figure 2.** The number of annual and perennial weeds per m$^{-2}$ in winter rye stands before harvesting in 2012–2016. Note: different letters indicate significant differences between the treatments ($p \leq 0.05$); FWR—field rotation with row crops; INT—intensive rotation; FGM—rotation for green manure; TC—three-course rotation; MONO—rye monoculture.

The main tillage of all crop rotations and monoculture is plowing, but, as pointed out by Rasmussen et al. [17], it cannot prevent severe invasions of perennial weeds. In other crop rotations, *Sonchus arvensis* (L.) Scop was the most common perennial weed. Its highest density was in the field with row crops in rye crop. In intensive crop rotation, *E. repens* was widespread, accounting for 37.5% of the total number of perennial weeds in the rye crop, while *Plantago major* L. accounted for 29.2%. *E. repens* and *E. arvense* accounted for the largest share of perennial weeds for the green manure rye crop rotation (both equal to 29.1%), but *E. repens* had a higher dry mass of 37.4%, while *E. arvense* accounted for only 26.4%. Scandinavian researchers studied the prevalence of *E. repens* in crop rotations with clover in the first year of use, which had been mowed four times, fertilized with manure or without it, and with or without catch crops. They argue that legumes and spring cereals caused the largest increase in weed population, especially when clover was grown before. Fertilization with manure and its incorporation by inoculation reduced the growth of *E. repens* by 28% [17].

Previous results of this study (since 2003) indicate that weed species found in the rye monoculture could be divided into six ecological groups according to soil pH demand, into five groups according to nitrogen demand, and into five groups according to soil moisture demand [34]. Weed species found in 2016 were divided into two ecological groups according to soil pH demand, three groups according to nitrogen demand, and three groups according to soil moisture demand. Thus, spraying herbicides on rye monoculture reduced the diversity of weed ecological groups. In 2003 rye monoculture, the highest abundance was moderately acidic, indifferent to nitrogen demand, moderately moist, and a moist-soil weed species [35]. In 2016, indifferent to soil pH demand, nitrogen, and moisture, as well as a moist soil weed species prevailed. These studies show that the species composition of weeds varies in monoculture.

Monocotyledonous weeds are more difficult to control in cereal crops. In particular, if the crop structure is dominated by *Poaceae* plants, *A. spica-venti* is widespread [36]. When winter rye was grown in crop rotations, monocotyledonous weeds were similarly widespread and did not differ significantly from monoculture, except for three-course crop rotation, where they were the least numerous. The results of other researchers [37] also show that long-term monocultures do not reduce the species composition of weeds. In the monoculture of our experiment, the abundance of monocotyledonous weeds was on average as much as eight times lower compared to the cultivated rye crops after different pre-sowing in different crop rotations. *A. spica-venti* was found in monoculture and fields with row crops. *A. spica-venti* was the most widespread in monoculture, accounting for up to 18.1% of the total number of weeds. Monocotyledonous weeds were least spread in a three-course crop rotation.

Many weeds have matured the seeds, but others, due to weed species competition and plant competitive relations, disappear before flowering [28,38]. In the study, in most cases, there were more weed species and the number of weeds found before harvest compared to their germination in early spring. Estimation of the number of weeds in pre-harvest winter rye crops of different crop rotations in 2012–2016 revealed that the highest number of weeds was found in the green manure crop rotation. It differed almost two-fold from the three-course crop rotation, where the number of weeds before the harvest was the lowest and differed significantly from the number of weeds in all the other crop rotations studied (Figure 2). The weediness of the rye crop in intensive crop rotation was very close to the one for the green manure crop rotation, while the perennial weeds were almost twice as few.

Assessment of the distribution of annual weeds showed that their statistically largest increase compared to the field with row crops and monoculture was in rye intensive crop rotation, but only in 2012. In later years, namely 2013–2016, the spread of weeds was weaker, as they could have been inhibited by growing catch crops for green manure (Figure 2). Researchers point to the benefits of catch crops—improved crop yields, resulting in higher water and nutrient uptake, increased weed control, and increased resistance to pests and diseases [14]. For the green manure crops and three-course crop rotations, annual weeds were distributed almost equally, no significant differences were found compared to other crop rotations. In the years 2012–2013, perennial weeds were most prevalent for the green manure crop rotation. A 3.7-fold difference was obtained by comparing the weediness of this crop rotation with the three-course crop rotation of the rye crop. Differences in the distribution of perennial weeds in 2013 were also determined between the rye crop and monoculture for the green manure and other crop rotations (2.1 times).

In 2013, the weed dry mass of monoculture was significant, 3.6 times higher compared to rye crops grown in crop rotations (Figure 3). Non-compliance with crop rotation and repeated reseeding of cereal encourages the spread of weeds specific to cereal. This was also determined by this study, with *M. arvensis* and *E. arvense* accounting for almost all of the weed dry mass in rye monoculture. The spread of characteristic weeds in crop rotations is a major phytosanitary factor that limits crop and soil productivity [14,17,38], and therefore the yield of monoculture was one of the lowest compared to other crop rotations. Under a drier vegetation period in 2014, the dry mass of weeds in rye crops was not affected by either monoculture or crop rotation. In 2014–2016, most weeds were detected in monoculture. Researchers who have conducted research on perennial winter wheat monoculture and different crop rotations also share our view that the main elements of sustainable agricultural activity are crop rotation with legumes, roots, and catch crops, as well as tillage using crop residue as a mulch [16,34,39]. The mass of weeds, especially perennial, increased significantly in 2016 for the green manure crop rotation in rye crop—3.6 times on average compared to the field with row crops, intensive, and three-course crop rotations. In the aforementioned crop rotation, rye was sown after rapeseed was added for the green mass. In previous years of observation, an increase in the mass of perennial weeds was also observed, although insignificant compared to other crop rotations. Zikeli S. and

Gruber S. [40] conducted their research on three organic farms with different tillage systems and plants for green manure. The researchers observed that growing and mulching perennial grasses for green manure creates favorable conditions for the spread of perennial weeds such as *Taraxacum officinale* F. H. Wigg. and *Vicia hirsuta* (L.) Gray., in particular, if it is applied for more than one year. Moreover, plant debris and the onset of perennial weeds hinder the preparation of a quality seedbed. For green manure crop rotation, winter rape also gains a lot of mass which does not mineralize before rye sowing and hinders the preparation of the seedbed. During the five years of the experiment, the majority (67.5%) of the total weed dry mass in crop rotations consisted of only three types: *E. repens*, *E. arvense*, *Cirsium arvense* (L.) Scop, and *M. arvensis* and *E. arvense* in monoculture in 2016. *T. perforatum* accounted for 47.2%, *A. spica-venti* 39.3%, and *C. regalis* 8.0%. of total short-lived weed mass in monoculture. Researchers argue that in their long-term experiments with cereal monoculture, aggressive and difficult-to-eradicate weeds *A. spica-venti* and *Papaver rhoeas* L. also prevailed [34].

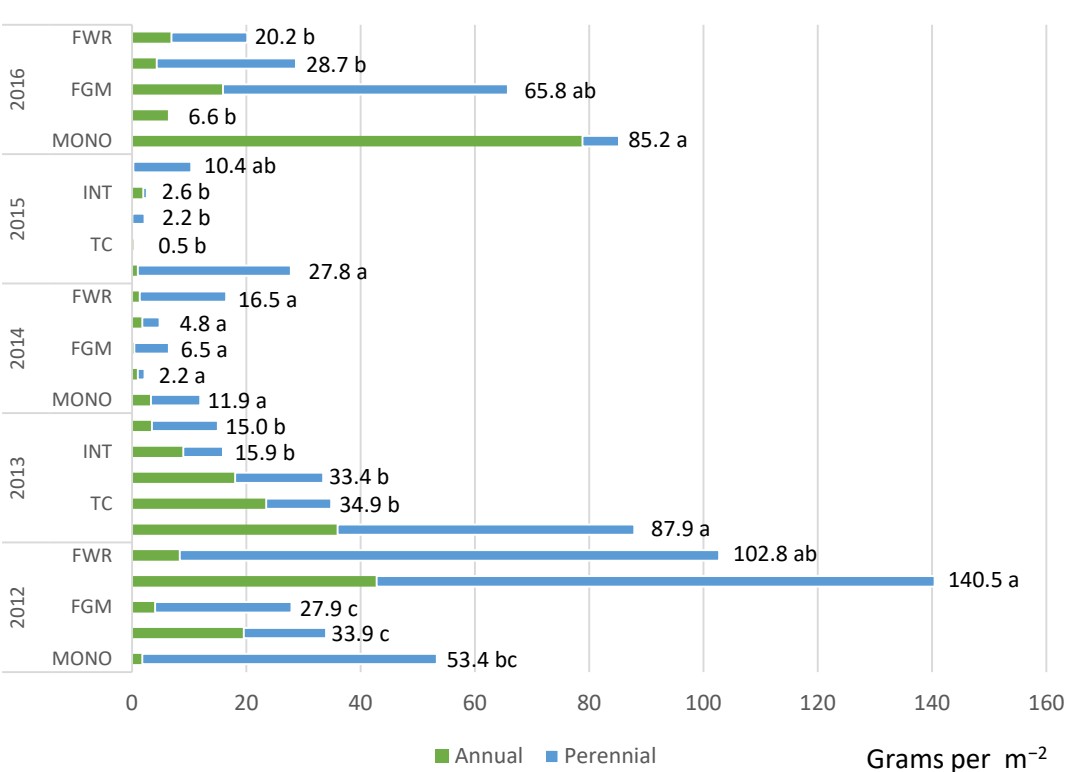

**Figure 3.** The mass of weeds in winter rye crop in different rotations before the harvest in 2012–2016. Note: different letters indicate significant differences between the treatments ($p \leq 0.05$); FWR—field rotation with row crops; INT—intensive rotation; FGM—rotation for green manure; TC—three-course rotation; MONO—rye monoculture.

A very strong correlation was found between the amount of precipitation during rye vegetation and the average weed mass in crop rotations $y = 7.98 \times 1.012$; $r = 0.91$; $p = 0.01$. When evaluating the influence of weed mass on rye yield and yield structure elements in 2016, a very strong statistically significant negative correlation was obtained. As the weed mass increased, the rye yield decreased: $y = 82.97 - 14.82x$, $r = -0.97$, $p \leq 0.01$. Additionally, with the increase of weed mass in the crop, the weight of 1000 rye grains decreased, and the grains shrank. A linear very strong negative correlation was obtained between these indicators: $y = 475.56 - 11.93x$, $r = -0.91$, $p \leq 0.05$. Researchers argue that higher crop infestation by weeds worsens plant health and depletes the soil, especially in monoculture, leading to lower grain yields [40]. Typically, grain yield decreases due to a decrease in the value of one or more yield components—number of productive stems, grain mass per ear of rye, the weight of 1000 grains, and the number of grains per ear of rye [41].

The results of subsequent studies were also similar, e.g., the cultivation of winter durum wheat in a perennial cereal monoculture significantly reduced the grain yield compared to their cultivation in crop rotations [16].

In 2012, 2014, and 2016, after researching the number of weed seeds in the cultivated soil of various crop rotations of rye crops, at a depth of 0–20 cm, the arable layer, from 20.2 to 71.4 thousand per m$^{-2}$ area weed seeds were found (Figure 4). During different tests of the weed seed bank implemented in 2002–2004 at the Vytautas Magnus University Experimental Station, the number of seeds found in the arable layer (0–25 cm) was 29.7–57.1 thousand [42] and 39.3–45.0 thousand [43]. The results show that the number of weed seeds in crops under conventional arable tillage system can vary both in different crops and in different crop rotations. The results of research conducted by He et al. [37] show that when assessing the weediness of long-term monocultures, the species composition of weeds in the soil bank in different crop fields becomes homogenized, regardless of the type of crop. This study showed that winter rye was dominated by the same annual weeds seeds all the years of the study—namely, *Chenopodium album* and *E. crus-galli*, but other types of weeds were also found: *Stellaria media, Fallopia convolvulus, J. bufonius, Sinapis arvensis, Lamium purpureum, Persicaria lapathifolia.* Perennial *M. arvensis* dominated only in monoculture. Most of the seeds were concentrated in the upper (0–10 cm) soil layer.

Evaluation of three-year data did not reveal significant differences in the number of weed seeds between different crop rotations every year. Three-year data showed that there were no significant differences in the number of weeds neither in the whole soil nor in different depths (Figure 4). They were not identified among different rye crop rotations in 2012 as well, but in 2014 and 2016, significantly fewer seeds were found in the three-course crop rotation compared to the intensive and green manure rye crop rotations, as well as with monoculture.

The results of Sjursen et al. [44] showed that the number and species composition of weed seeds often depend on the weediness of the crop in that year and the species composition of the predominant weeds [44]. Pilipavičius [45] found that the disintegration of weed seeds depends on the sum of active temperatures, precipitation, and duration of sunlight. His research has shown that with the increase in the sum of active temperatures, the spreading of weed seeds intensifies by up to 54%, with the increase in the sunshine duration by 14-51%, and with the increase of the precipitation, it decreases by 16–57%. Pupalienė et al. [46] indicate that a study carried out in 2008 and 2009 showed a very strong, statistically significant relationship between weed abundance in the crop and the number of weed seeds in the soil. In this experiment, the same tendency was also observed in rye crops i.e., with higher weediness before the harvest, more weed seeds are usually detected in the soil, but no statistically significant dependence was found.

The average results of 2012–2016 show that in some years, the cultivation of winter rye in monoculture significantly reduced yield compared with the rye yield grown in crop rotations (Figure 5). When growing rapeseed, for green manure crop rotation, rye yield also decreased significantly in unfavorable years.

In all the years of the experiment, rye yielded best in the three-course field with row crops, and intensive crop rotations. Woźniak [16] conducted an experiment in monoculture, crop rotation, and three tillage systems, and concluded that grain yield and formation of productivity elements were more influenced by crop rotation than by tillage system. However, productivity was still lower in monoculture compared to wheat crop rotation.

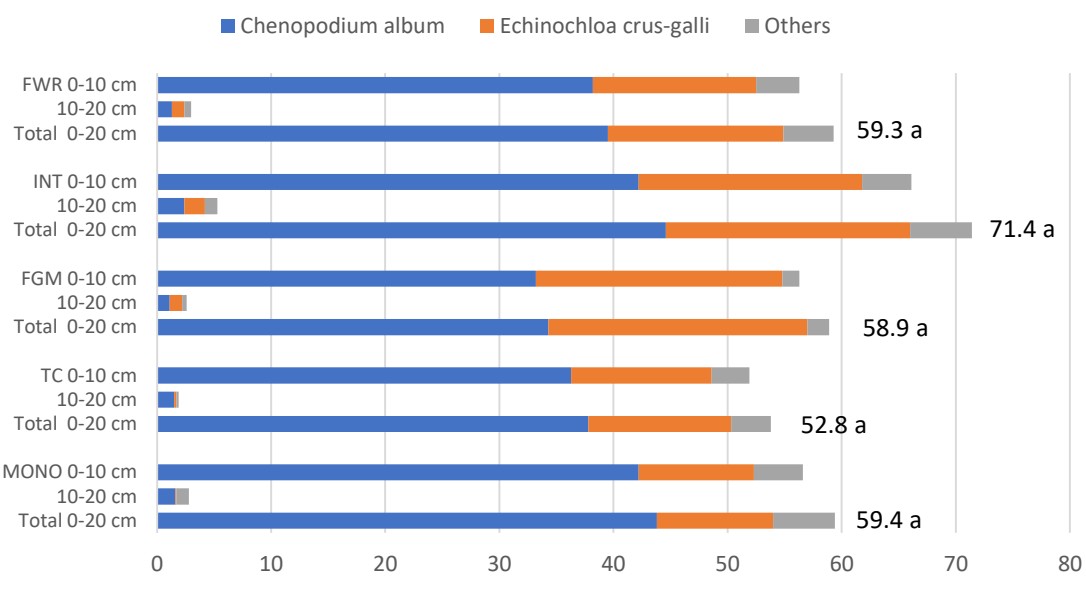

(**a**)

(**b**)

**Figure 4.** *Cont.*

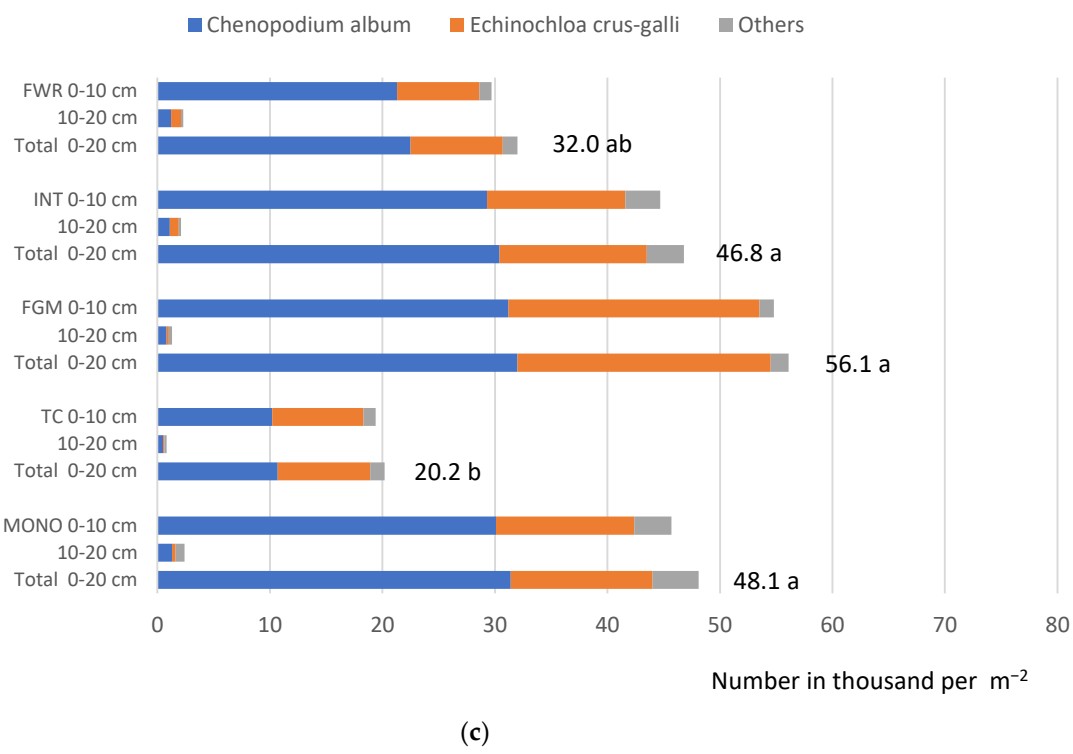

(**c**)

**Figure 4.** Weed seed bank dominant species of weed seeds after winter rye harvest in 0–10 and 10–20 cm soil layers in (**a**) 2012; (**b**) 2014; (**c**) 2016. Note: different letters indicate significant differences between the treatments ($p \leq 0.05$); FWR—field rotation with row crops; INT—intensive rotation; FGM—rotation for green manure; TC—three-course rotation; MONO—rye monoculture.

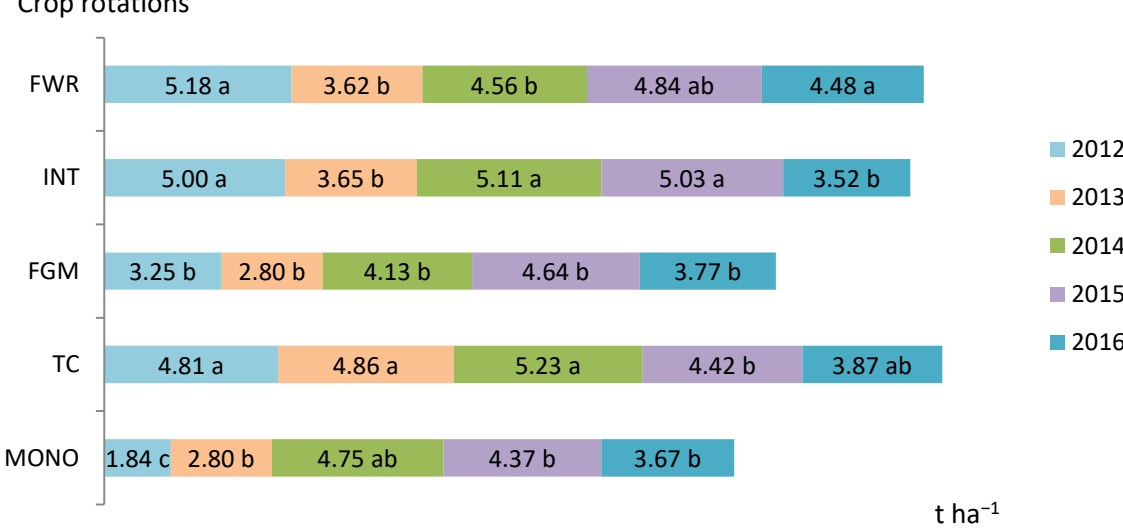

**Figure 5.** Winter rye grain yield in different crop rotations, 2012–2016. Note: different letters indicate significant differences between the treatments ($p \leq 0.05$); FWR—field rotation with row crops; INT—intensive rotation; FGM—rotation for green manure; TC—three-course rotation; MONO—rye monoculture.

## 4. Conclusions

Summarizing the average results of five years of investigation shows these conclusions:

1. Meteorological conditions influenced the weed amount and mass, and species like *T. perforatum* in 2016. A very strong correlation was found between the amount of precipitation during rye vegetation and the average weed mass in crop rotations ($r = 0.91$; $p = 0.01$). The results of the five-year investigation show that in the rye stand, the lowest weed abundance was found in three-course crop rotation with bare fallow compared to other rotations. In fertilized and herbicide-sprayed monoculture, the density of weeds before the harvest was the lowest compared with the field rotation with row crops, intensive, and the green manure rotations, but the weed dry mass was the highest in monoculture.

2. Monoculture leads to the spreading of weeds such as *M. arvensis*. and *E. arvense*, which are difficult to control. There was also an increase in the spread of *A. spica-venti*, especially in monoculture and in the field rotation with row crops.

3. As the weed density in the rye stands increased, the weight of 1000 grains decreased. Among these indicators, a linear very strong negative correlation was obtained: $r = -0.98$, $p \leq 0.01$. As the weed dry mass increased, rye yields decreased. Among these variables, a linear significant very strong negative correlation was obtained: $r = -0.97$, $p \leq 0.01$. Additionally, as the weed mass of the crop increased, the weight of 1000 rye grains decreased. Among these indicators, a linear very strong negative correlation was obtained: $r = -0.91$, $p \leq 0.05$.

4. Weed seed bank was found to be the lowest in the arable layer in the three-course crop rotation, dominated by the same annual weeds seeds in all the years and in all investigated crop rotations of the study—namely, *C. album* and *E. crus-galli*. Seeds of *M. arvensis* were established only in monoculture. Most of the seeds were concentrated in the upper (0–10 cm) soil layer. Crop rotations with perennial grasses help to prevent the spread of monocotyledonous weeds, especially *E. crus-galli*.

**Author Contributions:** L.M.B., R.P. and I.A. designed the research framework and contributed to the application of the study methodology and the analysis of the results. V.S., L.S. and V.B. played an active role in writing, reviewing and editing the manuscript. All authors have read and agreed to the published version of the manuscript.

**Funding:** This research received no external funding.

**Institutional Review Board Statement:** Not applicable.

**Informed Consent Statement:** Not applicable.

**Data Availability Statement:** Not applicable.

**Conflicts of Interest:** The authors declare no conflict of interest.

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
