# Peer review of "The Influence of Long-Term Different Crop Rotations and Monoculture on Weed Prevalence and Weed Seed Content in the Soil"

_agronomy, doi:10.3390/agronomy11071367_

Round 1

Reviewer 1 Report

The manuscript presents the results of a study about the effects of crop rotation Vs monoculture on weed density and weed soil seed bank. The subject of the study is undoubtedly important and the results of a long-term experiment are valuable for crop and weed science. However, the quality of the manuscript needs major revision for publication.

Throughout the text, English language has to be revised. Appropriate terminology must be used. In some paragraphs, text formatting is also an issue. Full scientific names of species are usually written the first time the species is mentioned and shortened names are used further in the text.

Some specific question and comments:

Ln 102 The reported results refer to 2012-2016, so would it be possible to draw any conclusions about the effects of climate change?

Ln 203 Were 20 samples taken from each experimental plot?

Ln 214 The formula for calculating the number of weed seeds per area includes depth (thickness) but not the sampling area.

Ln 221 what were the factors in the ANOVA?

The type of the figures 1, 2 and 5 is not appropriate for the kind of data shown. If each year was used as a repetition (replication?) shouldn’t the average vales be reported? Including blocking factor (individual plot) into the statistical model might be necessary. The results of the statistical analysis must be reported in the text (not just in the figures).

Ln 251-252 The sentence is hard to understand. What is “crop rotation of Echinochloa crus-galli”? Were the plots with the intensive crop rotation more moist that the others?

Ln 257-258 Was this more an effect of the year and the plot (e.g. soil seed bank), than of the crop rotation? The conclusion made in the next sentence is too general.

Ln 305-314 Were herbicides not used before 2012? It is not possible to determine, whether there was any effect of herbicides or the effect of the monoculture.

Ln 315-316 I’m not sure that this is relevant, crop seeds are not supposed to survive and spread in the environment.

Ln 330 What are “interspecies”?

The Results and Discussion section should be separated into two sections. As it is, the section is not well structured and is difficult to read.

Ln 378 Taraxacum and Vicia are not perennial grasses

Ln 411-412 Is the weed seed number reported per unit area, or soil volume, or soil mass?

Ln 430-432 species composition of the weed soil seed bank depends on weed species composition in the field (vegetation)?

Ln 435 what is “falling of weed seeds”?

Ln 451-452 What was the yield in the monoculture compared to? Again, the average values should be reported. “The yield of monoculture was equal to the cultivation of cereal in crop rotation” – what does it mean?

Ln 459 “In all the years of the experiment, rye yielded best in intensive, three-course and field 459 with row crops rotations” – according to Fig.5, this was not the case?

One of the most important factors influencing weed density and composition is the pre-crop, however, it was not analyzed. Crop competitiveness may also be important (why were there more weeds in green manure crops?).

It is not surprising that the seeds in the soil were concentrated in the top layer of the soil, as sampling was done before tillage.

It is an important finding, that the soil seed bank was reduced after three-year rotation, more analysis is needed.

The effect of crop rotation on particular problematic weed species could be analyzed and shown more specifically (Apera spica-venti, Elytrigia repens, Echinochloa crus-galli etc).

Conclusions are written very poorly and must be improved. No results should be reported in the Conclusions.

Author Response

[2021-06-25]

Dear Reviewer:

We wish to submit an original research article for publication in Agronomy. We have made significant improvements to this article, titled “The influence of long-term different crop rotations and monoculture on weed prevalence and weed seed content in the soil”.

We would like to sincerely thank You for your insightful comments and recommendations that allowed us to fundamentally correct the article.

Comments and Suggestions for Authors

The manuscript presents the results of a study about the effects of crop rotation Vs monoculture on weed density and weed soil seed bank. The subject of the study is undoubtedly important and the results of a long-term experiment are valuable for crop and weed science. However, the quality of the manuscript needs major revision for publication.

Throughout the text, the English language has to be revised. Appropriate terminology must be used. In some paragraphs, text formatting is also an issue. Full scientific names of species are usually written the first time the species is mentioned and shortened names are used further in the text.

Some specific question and comments:

Ln 102 The reported results refer to 2012-2016, so would it be possible to draw any conclusions about the effects of climate change?

The first sentence of the first conclusion is about the influence of meteorological conditions. And we add a sentence about correlation.

Ln 203 Were 20 samples taken from each experimental plot?

Samples were taken from each experimental plot (from all replications) from 20 places using an agrochemical drill (Ø 20 mm), an average sample from experimental plot was compound.

Ln 214 The formula for calculating the number of weed seeds per area includes depth (thickness) but not the sampling area.

It is used soil density in the formula and it is calculated the mass of soil of 0-20 cm layer.

Ln 221 what were the factors in the ANOVA?

One-factor ANOVA (the factor – crop rotation)

The type of figures 1, 2, and 5 is not appropriate for the kind of data shown. If each year was used as a repetition (replication?) shouldn’t the average values be reported? Including blocking factors (individual plots) into the statistical model might be necessary. The results of the statistical analysis must be reported in the text (not just in the figures).

Ln 251-252 The sentence is hard to understand. What is “crop rotation of Echinochloa crus-Galli”? Were the plots with the intensive crop rotation moister than the others?

It was corrected.

Ln 257-258 Was this more an effect of the year and the plot (e.g. soil seed bank) than of the crop rotation? The conclusion made in the next sentence is too general.

It is an effect of crop rotation - Consolida regalis Gray did not spread in other crop rotations and was not observed in any other years.

Ln 305-314 Were herbicides not used before 2012? It is not possible to determine, whether there was any effect of herbicides or the effect of the monoculture.

Agrotechnical measures in monoculture are used as in other crop rotations. There are cited results from the experiment 10 years before.

Ln 315-316 I’m not sure that this is relevant, crop seeds are not supposed to survive and spread in the environment.

Weed seeds are better adapted to spread and survive in the environment compared to agricultural crops.

Ln 330 What are “interspecies”?

It was corrected.

The Results and Discussion section should be separated into two sections. As it is, the section is not well structured and is difficult to read

There is no strict requirement for two separated sections.

Ln 378 Taraxacum and Vicia are not perennial grasses

We thank the Reviewer for the comment. They are weeds, not grasses. Taraxacum is a perennial weed. Vicia hirsuta – annual weed.

Ln 411-412 Is the weed seed number reported per unit area, or soil volume, or soil mass?

It is reported per unit area – 1 m2.

Ln 430-432 species composition of the weed soil seed bank depends on weed species composition in the field (vegetation)?

Yes, it depends on weed species growing in the field.

Ln 435 what is the “falling of weed seeds”?

We thank the Reviewer for the comment. It was corrected. Spreading from weeds growing in the field.

Ln 451-452 What was the yield in the monoculture compared to? Again, the average values should be reported. “The yield of monoculture was equal to the cultivation of cereal in crop rotation” – what does it mean?

We don’t use averages of all experimental years.

It means: There are no significant differences between yields of rye grown in monoculture compared with rye grown in crop rotations.

Ln 459 “In all the years of the experiment, rye yielded best in intensive, three-course and field with row crops rotations” – according to Fig.5, this was not the case?

In many cases.

One of the most important factors influencing weed density and composition is the pre-crop, however, it was not analyzed. Crop competitiveness may also be important (why were there more weeds in green manure crops?).

There are different pre-crops in crop rotations. It is a very old experiment. Crop competitiveness may also be important.

It is not surprising that the seeds in the soil were concentrated in the top layer of the soil, as sampling was done before tillage.

Yes.

It is an important finding, that the soil seed bank was reduced after a three-year rotation, more analysis is needed.

Thank you for your consideration. Black fallow is applied in three-course crop rotation, where weeds were destroyed and could not mature seeds. For this reason seeds bank was reduced in three-course crop rotation.

The effect of crop rotation on particular problematic weed species could be analyzed and shown more specifically (Apera spica-venti, Elytrigia repens, Echinochloa crus-galli etc).

It can be the topic of other paper.

Conclusions are written very poorly and must be improved. No results should be reported in the Conclusions.

We thank the Reviewer for the comment, but there are different opinions about the results reported or not reported in conclusions.

We reviewed and clarified the manuscript. Thank you for your consideration.

Sincerely,

Lina Skinulienė

Vytautas Magnus University, K. Donelaičio str. 58, 44248 Kaunas, Lithuania

[+37067412525]

[email protected]

Reviewer 2 Report

The manuscript evaluates the effect of different 5-year rotations on weed densities and seedbank dynamics in a long-term experiment. The results are important in highlighting the effect of diverse rotations in reducing the weed pressure. Paper is well written and technically sound; however, I suggest using tables for bigger datasets instead of stacked bar charts.

Author Response

[2021-06-25]

Dear Reviewer:

We wish to submit an original research article for publication in Agronomy. We have made significant improvements to this article, titled “The influence of long-term different crop rotations and monoculture on weed prevalence and weed seed content in the soil”.

We would like to sincerely thank You for your recommendations, but there are different opinions about using tables for bigger datasets so we used stacked bar charts.

Sincerely,

Lina Skinulienė

Vytautas Magnus University, K. Donelaičio str. 58, 44248 Kaunas, Lithuania

[+37067412525]

[email protected]